# The role of customer experience in the effect of online flow state on customer loyalty

**Adnan Veysel Ertemel[1,2], Mustafa Emre Civelek[1], Güzide Öncü Eroğlu Pektaş[2]\*, Murat Çemberci[3]**

1 International Trade Department, Istanbul Commerce University, Istanbul, Turkey, 2 Maritime Transportation Management Engineering, Istanbul University-Cerrahpaşa, Istanbul, Turkey, 3 Department of Business Administration, Istanbul Yıldız Technical University, Istanbul, Turkey

\* guzide.pektas@iuc.edu.tr

**Data Availability Statement:** All relevant data are within the paper and its S1 Data file.

**Funding:** The author(s) received no specific funding for this work.

## Abstract

### Purpose

The Internet revolution has radically changed the means of conducting business all over the world in the past few decades. The digital medium enables consumers worldwide to shop online through B2C e-commerce websites in a convenient manner. Online websites compete to provide a compelling and seamless brand experience to retain their customers. In order to achieve this, fostering a state of flow may help the brands increase customer experience, customer satisfaction and loyalty. In this study, the aforementioned phenomenon is tested against Turkish university students.

### Methodology

The study was conducted against 538 valid respondents. The results of the survey were interpreted with the structural equation modeling method. Quantitative data were obtained using a five-point Likert scale. Initially, confirmatory factor analyses and reliability analysis were performed, respectively in order to determine the validity and reliability of the scale.

### Findings

As a result of the analyses, it has been empirically proven that an online flow state, which is a momentary phenomenon, helps online e-commerce websites build customer satisfaction and customer loyalty indirectly through customer experience. These results are partly parallel with those in the extant literature.

### Originality

This study is significant in the literature in that, as opposed to the extant literature, online flow state is found to influence customer satisfaction and customer loyalty rather indirectly via moderating effect of customer experience. Additionally, it is the first to incorporate customer satisfaction along with customer loyalty as a new construct affected by online flow state and customer experience. The results also have important managerial implications.

**Competing interests:** The authors have declared that no competing interests exist.

## 1. Introduction

The highly competitive new marketing environment of the 21st century calls for new approaches for marketers for appealing to their customers. This is because consumer behavior is evolving at an unprecedented pace. The Internet is an evolving technology, which has facilitated the development of new business relationships among brands and their customers [1–4]. The brands now have a much more convenient way to reach their consumers and benefit more in online context [5].

As today's fragmented, highly digitized marketplaces have become more complex than ever, engaging new generation consumers have become more of a concern. Therefore, many of the old marketing strategies turn out to be ineffective and new marketing strategies emerge that appeal rather to the unconscious. These strategies require consumers to spend minimum mental effort, which makes them feel in a flow state [6], get entertained without realizing how the time passes by [7] and focus more on the experiential aspects of consumption [8]. The aforementioned strategies are so effective in engaging consumers that they can even result in addictive behaviors [9–11]. As such, these marketing strategies enable the marketers to foster higher levels of involvement [12]. In terms of attaining this goal, flow theory is an important theory that enables the marketers to achieve a high level of involvement, especially in an online setting. In this regard, e-commerce is a perfect medium that enables consumers to order their necessities of all assortments with just a few clicks. Furthermore, the recent covid-19 pandemic has further accelerated the adoption of e-commerce worldwide. Over the last years, the importance of the online flow state has been emphasized to build a better customer experience.

In this study, the youth segment was chosen as the target group to measure the flow experience in e-commerce use in Turkish consumers' consumption patterns. This particular sample group was chosen because of the high potential of this group and the higher mobile usage rates of young people In Turkey. Even though online flow is a popular concept studied in various contexts, the effect of online flow state on the longer-term phenomenon is not probed in the extant literature except for the one done in a narrow scope by Shim, Forsythe, Kwon [13]. Considering this fact, this study makes a unique contribution to the literature by incorporating both customer satisfaction and customer loyalty as being affected by online flow state directly or indirectly through brand experience.

## 2. Conceptual background

In this study, the relationship among online flow state, customer satisfaction, customer experience and customer loyalty are analyzed using Structural Equation Modeling (SEM) method.

The initial model is depicted in Fig 1.

### 2.1. Flow state

Flow state is defined as a fully immersed state experienced when someone is totally involved in an activity [6]. It is such a state that nothing else seems to matter to the individual [14] and causes the individual to experience a loss-of-control and centering of attention at the same time. Optimum flow state, also known as autotelic experience is the state in which the individual's skills are fully involved in overcoming a significant but manageable challenge [15].

**2.1.1 Online flow state.** Online flow state, on the other hand, can be defined as a totally absorbing, fully engaging online experience state [16, 17]. It can be described as a multidimensional construct that encompasses sense of being in control, intrinsic enjoyment, sense of time distortion and tele-presence [18]. Online flow can occur in various online activities including but not limited to e-commerce [19], e-learning environments and online gambling [18]. In the e-commerce context, online flow can be seen as the extent to which consumers are engaged in

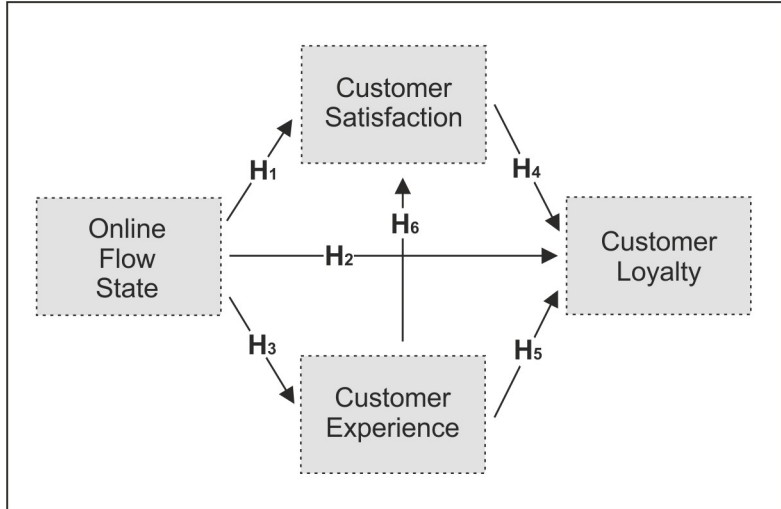

**Fig 1. Theoretical model.**

interacting with the brand-related stimuli [13]. As it becomes harder for the brands to engage new generation consumers, the importance of online flow in customer experience becomes more prevalent. Various studies on online flow in recent years can be shown as an indicative of increased attention on the phenomenon [9, 12, 13, 18].

## 2.2. Customer satisfaction

Customer satisfaction is defined as a customer's overall judgment on disconfirmation between the expected and perceived service performances [20]. If the perceived performance meets or exceeds the expectation, the customer is satisfied; otherwise, the result will be dissatisfaction [21].

Customer satisfaction is a transaction-specific measure, which means that a customer evaluates his/her perception of performance relative to expectation in each service encounter, independently of the other occasions [22–26].

Satisfying their customers is one of the ultimate goals that e-businesses seek due to the long-term benefits of having satisfied customers such as positive word of mouth comment, customer loyalty, and sustainable profitability [27–29].

Customer satisfaction is a critical factor for customer loyalty and customer satisfaction [30]. The prerequisite of customer loyalty is to ensure satisfaction. According to some researchers, e-customer satisfaction is due to website features. In this perspective, it has been found that concrete elements, responsiveness, interaction and stability have a significant impact on customer satisfaction in the online context [31]. In today's highly competitive online marketplace, high level of service performance is a differentiator in competition, and an effective way to improve customer satisfaction and loyalty [32]. In order to obtain high level of customer satisfaction, high service quality is needed, which often leads to favorable behavioral intentions [33].

## 2.3. Customer experience

Customer experience is the subjective feeling that stays after the user purchases a product or service, aiming to manage the processes of experiences as perceived by customers in their relationship with the brands [34]. Brands reinforce these subjective feelings to activate their

customers' five senses in their efforts to enhance customer experience [35]. Customer experience is of critical importance in various sectors including hospitality [32] and online services [36–38]. Chase and Dasu [39] suggest that the sole reason for employment of behavioral science by brands is to enhance the customer experience.

When a customer buys an experience, he/she pays money for a series of events to spend enjoyable time, which will stick in mind just like a play performed on stage of a theatre [40]. Some authors argue that customer experience will be the next competitive battlefield for businesses [41]. Both online and offline consumers are looking at five categories when shopping, namely; location, convenience, knowledge, personality and price [42]. Studies conducted by Nisar and Prabhakar [43] and the one conducted by Gentile, Spiller, Noci [44] pointed out that, the sensory, emotional, cognitive pragmatic, lifestyle and relational situations of the customer affect the experience. All of these factors cause the customer experience to be perceived differently by consumers.

## 2.4. Customer loyalty

Customer loyalty is defined as a customer's attitude to the service [22, 33, 45]. In addition, it is formed by a customer's cumulative experience with the service over time, not by a specific service encounter [20, 46–49]. Customer loyalty is very essential to the organization in order to retain its current customers [50, 51].

Loyalty of a firm's customer has been recognized as the dominant factor in a business organization's success. [52]. Customer loyalty is very essential to the organization in order to retain its current customers [51]. Loyalty is mainly expressed in terms of revealed behavior [53]. Customer's repurchase behavior is estimated as a basic requisite for loyalty that is followed by satisfaction [54]. By understanding the importance of customer loyalty, an organization can build commitment by having existing customers re-purchase their products and services [55–57].

## 3. Hypothesis development and research model

### 3.1. The relationship between flow state and customer satisfaction

Many studies in the extant literature shows a significant positive influence of flow state on customer satisfaction in various contexts including hospitality [58, 59] and sports [60, 61] industries. Hoffman and Novak [18] had also compiled the results of various studies on the effect of the flow state on other constructs. Kim and Han [14] found out that customers understand and enjoy mobile marketing messages more as they are fully absorbed and totally focused on flow state, and that this high involvement flow state facilitates purchase intentions including but not limited to customer satisfaction.

Thus, in the light of the extant literature, we can hypothesize that;

$H_1$: *Flow State has a positive effect on Customer Satisfaction*

### 3.2. The relationship between flow state and customer loyalty

Since an online flow state represents an optimal state that is joyful and entertaining, it is meaningful to evaluate it as a phenomenon that ultimately leads to customer loyalty. Hausman and Siekpe [16] found that flow affects online consumers' return intention and thus loyalty. Zhou, Li, Liu [62] found that online flow has a significant effect on users' loyalty in mobile social networking sites. Smith and Chen [63], on the other hand, indicate that brand experience sub constructs affect branding efforts, which paves the way to brand loyalty.

Previously, Luna, Peracchio, de Juan [64] found out that online flow experience could lead to 'sticky' web sites. Sticky, in this context, means that the website captures consumers' attention in such a way that consumers spend prolonged periods of time on the site because of the compelling nature of the experience [18].

Likewise, Bilgihan [65] indicated optimal flow state as an important precedent to loyalty in the e-commerce environment. Although online flow state is a momentary, rather short-term experience, some scholars argue that it also helps improve brand experience and customer loyalty in the long term [13].

Thus, we can hypothesize that;

**H₂**: *Flow State has a positive effect on Customer Loyalty*

## 3.3. The relationship between flow state and customer experience

Schembri [66] pointed out that online flow contributes to the experiential meaning of a brand ultimately enhancing customer brand experience. Similarly, Shim, Forsythe, Kwon [13] and Müller, Flores, Agrebi, Chandon [67] suggests that unlike other traditional channels, a brand's website can deliver an interactive, optimal and extraordinary flow experience that ultimately helps create a positive overall customer experience.

Shim, Forsythe, Kwon [13] argues that the reason behind this can be explained by the fact that the online flow state is related to all sensory, intellectual, behavioral and affective dimensions of the brand experience. For example, online flow can be understood as a state where the individual is fully concentrated, implying that all visual and auditory senses are highly active in this state. This intense state of mind also triggers brain activity that implies the intellectual brand experience.

Additionally, the tele-presence dimension of online flow state implies an effect on behavioral brand experience. Lastly, autotelic experience in online flow state helps the building of effective brand experience.

Therefore, we can hypothesize that;

**H₃**: *Flow State has a positive effect on Customer Experience*

## 3.4. The relationship between customer satisfaction and customer loyalty

Many studies have been undertaken to date to examine the relationship between customer satisfaction, service performance and customer loyalty in a variety of service industries, including tourism [68, 69], *medical* [70, 71], *and telecommunications* [26, 72] industries. The majority of these studies have found substantial causal links between service performance, customer satisfaction, and customer loyalty [26]. The opportunity to bond with consumers and the realization of a brand's emotional characteristics can overcome the pervasive instability that exists in online environments [73, 74]. Brand loyalty can decrease switching behavior and increase consumer retention rate [75]. Brand loyalty has also been linked to repeat purchase behavior [63].

Repeat purchase behavior helps the realization of loyalty. If the customer prefers that particular brand even if there are similar brands available, then this demonstrates that loyalty has been established. In order for loyalty to be established, the recurring customer satisfaction should take place without any exceptions and be free of bad experience.

Hence, in the light of the extant literature, we can formulate the following hypothesis;

**H₄**: *Customer Satisfaction has a positive effect on Customer Loyalty*

### 3.5. The relationship between customer experience and customer loyalty

Considering consumer brand loyalty is generally established on the basis of long-term and close interactions between a customer and a brand, previous studies [76–78] have revealed a link between customer experience and customer loyalty. Some studies indicate that a positive customer experience can greatly boost brand loyalty [73, 79]. Lin and Kuo [80] found that consumers' loyalty intentions are affected by their recent purchases, suggesting that a positive brand customer experience may be the key to strong customer loyalty.

Brakus [81] conceptualized customer experience as a multidimensional construct and suggest that all kinds of customer experiences have the potential to affect customer loyalty.

Thus, we can formulate the following hypothesis.

$H_5$: *Customer Experience has a positive effect on Customer Loyalty*

### 3.6. The relationship between customer experience and customer satisfaction

The extant literature suggests that a superior customer experience helps build customer satisfaction [81, 82]. This phenomenon is also verified in e-commerce context by previous studies [73, 83]. Schmitt, Brakus, Zarantonello [84] further indicates that the strength of customer experience could affect customer satisfaction.

Thus, in the light of the we can formulate hypothesis.

$H_6$: *Customer Experience has a positive effect on Customer Satisfaction*

## 4. Research methods

Quantitative data were collected by means of the questionnaire designed in a five-point Likert scale. Firstly, confirmatory factor analyses and reliability analysis were conducted to determine the validity and reliability of the scale. Structural Equation modeling as a multi variable statistical technique was employed to test the hypotheses of the theoretical model [85].

This technique was used to understand the indirect and direct effects in the theoretical model [86] and to decrease measurement errors [87]. The analyses were performed with SPSS and AMOS statistics programs.

### 4.1. Measures and sampling

The scales taken from previous research measured the dimensions in the initial model of the study. The Likert scale in 5-point was used from a strong disagreement to strong acceptance.

The questionnaire was distributed to more than 700 individuals residing in Turkey via an online from. The survey was conducted in Turkish, among university students residing in 7 biggest cities in all geographical regions in Turkey, namely; İstanbul, İzmir, Antalya, Samsun, Ankara, Erzurum, Diyarbakır. The online questionnaire was accepted only from those adults with their explicit consent and who had previously purchased online in the past 12 months. 538 valid questionnaires from individuals were collected. 334 of the respondents were female and the remaining 204 respondents were male. In order to measure flow state, the scale suggested by Bilgihan, Okumus, Nusair, Bujisic [88] with 8 questions was used. In order to measure customer satisfaction, the scale suggested by Oliver [89] was used. The scale suggested by Brakus [81] was utilized to measure customer experience. Finally, the scale developed by Yoo and Donthu [90] was employed to measure brand loyalty.

## 4.2. Construct validity and reliability

At the beginning, exploratory factor analysis (EFA) was used for the data purification process. So as to understand convergent validity, confirmatory factor analysis (CFA) was utilized. This analysis was applied on the remaining 13 items [91]. The findings of the CFA determined the fit of the structural model. The Likelihood Ratio Chi-Square Test shows compliance with the original model and the acquired model [92]. $\chi2/DF$ was found as 2.856. This $\chi2/DF$ ratio is under the limit point of 3. Additionally, other fit indices also show acceptable results (i.e. CFI = 0.959, IFI = 0.959, RMSEA = 0.059).

Table 1 shows the CFA Results. As provided in the table, standardized factor loads for each item are obtained as significant (larger than 0.5). Average variance extracted values were near or above the limit point (i.e. 0.5) [87].

These results proved the convergent validity of the constructs. To appraise discriminant validity, the square roots of AVE values of each variable were obtained. In Table 2, the diagonals indicate the square root of AVE values. The reliability of each structure was calculated separately. Composite reliability and Cronbach α values are near or more than the limit point which is recommended as 0.7 [93].

Descriptive statistics of the dimensions, Cronbach α and composite reliabilities, average variance extracted values and Pearson correlations among the dimensions are presented in Table 2.

## 4.3 Test of the hypotheses

Maximum likelihood estimation method was utilized to test the hypotheses. It is the main estimation method of covariance-based structural equation modeling (CB-SEM). CB-SEM is a confirmatory method [86]. Therefore, in this research, it is used to confirm the hypotheses, which are developed by depending upon the base theories. To assess the structural model, the goodness of fit indices were utilized.

The absolute goodness of fit indices are the root mean square error of approximation (RMSEA) and the $\chi2$ goodness of fit statistic.

Note: $\chi2/DF$ = 2.356, CFI = 0.970, IFI = 0.970, RMSEA = 0.050

**Table 1. Confirmatory factor analysis results.**

| Variables | Items | Standardized Factor Loads | Unstandardized Factor Loads |
|---|---|---|---|
| **Customer Satisfaction (CSA)** | CSA0222 | 0.495 | 1 |
| | CSA0525 | 0.738 | 1.174 |
| | CSA0121 | 0.764 | 1.176 |
| | CSA0323 | 0.814 | 1.240 |
| | CSA0626 | 0.785 | 1.273 |
| **Customer Experience (CEX)** | CEX0715 | 0.571 | 1 |
| | CEX1018 | 0.756 | 1.211 |
| | CEX1220 | 0.579 | 0.964 |
| **Customer Loyalty (CLY)** | CLY0127 | 0.647 | 1 |
| | CLY0228 | 0.793 | 1.172 |
| | CLY0329 | 0.787 | 1.194 |
| **Flow State (FLS)** | FLS0404 | 0.623 | 1 |
| | FLS0505 | 0.776 | 1.126 |

p<0.01 for all items.

**Table 2. Construct descriptives, reliability and correlation.**

| Variables | 1 | 2 | 3 | 4 |
|---|---|---|---|---|
| 1. Customer Satisfaction | (.728) | | | |
| 2. Customer Experience | .589* | (.641) | | |
| 3. Customer Loyalty | .465* | .468* | (.745) | |
| 4. Flow State | .253* | .306* | .149* | (.703) |
| Composite reliability | .846 | .673 | .788 | .660 |
| Average variance ext. | .530 | .411 | .556 | .495 |
| Cronbach α | .826 | .683 | .783 | .657 |

*$p < 0.05$.

Note: Values in diagonals are the square root of AVEs.

The relative goodness of fit indices are incremental fit index (IFI) and comparative fit index (CFI). As Fig 2 shows, fit indices structural regression model satisfactorily determines fit of the model. χ2/DF value is 2.356 and between limit points (i.e. between 2 and 5). CFI is 0.970, IFI is 0.970. RMSEA is 0.050. These are adequate values. The results of the hypothesis tests are summarized in Table 3.

## 5. Analysis results

$H_1$ hypothesis is not supported. This means FLS does not have a direct effect on CSA. $H_2$ hypothesis is not supported. This means FLS does not have a direct effect on CLY. $H_3$ hypothesis is supported, which means that FLS has a direct effect on CEX. $H_4$ hypothesis is supported. This indicates that CSA has a direct effect on CLY. $H_5$ hypothesis is supported. This means that CEX has a direct effect on CLY. $H_6$ hypothesis is supported, which means that CEX has a direct effect on CSA.

The results indicate that online flow state does affect brand experience directly, and customer loyalty and customer satisfaction indirectly through the mediating effect of brand experience.

## 6. Discussion

The related studies on the effect of flow state on customer satisfaction find a direct, positive influence [59–61]. The analysis results of this study also find such a positive relationship according to the Pearson correlation table. However, the SEM analysis results indicate that there is an indirect effect of the flow state on customer satisfaction through the mediating role of customer experience. Thus, this study makes a significant contribution to the literature by highlighting this relationship in online context in e-commerce environment. Based on these results, it can be implied that online flow state, which is a momentary experience–is one that occurs at a specific time, does not directly affect a long-term phenomenon like customer loyalty and customer satisfaction. However, since online flow state results in a better overall brand experience, it does help build better customer loyalty and satisfaction through improved total brand experience. This result has significant managerial implications. Online website visitors have ever increasing expectations of being immersed with exceptional experiences that let them have a feeling of distortion in time and space [15] without requiring them to think about unnecessary details [7]. This is especially true in today's highly fragmented, fairly busy daily lives characterized by the attention economy [94]. In such an environment, consumers are being bombarded with various stimuli coming from all sources and thus have difficulty paying their scarce and valuable time and attention to any of those stimuli. Therefore, online flow

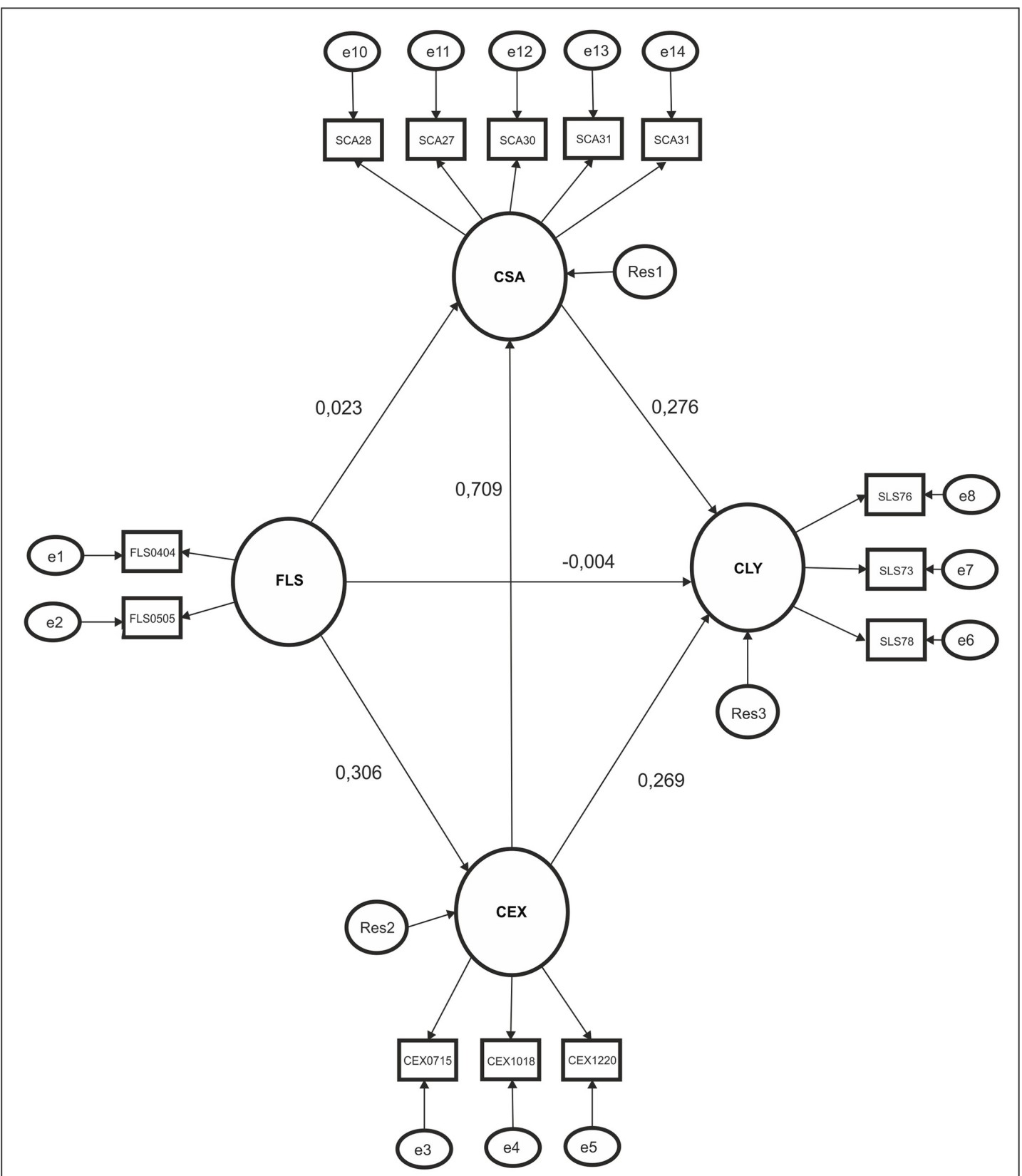

**Fig 2. Results of the SEM analysis.**

**Table 3. Hypotheses test results.**

| Relationships | Standardized Coefficients | Unstandardized Coefficients | Hypotheses | Results |
|---|---|---|---|---|
| FLS → CSA | 0.023 | 0.018 | $H_1$ | Not Supported |
| FLS → CLY | -0.004 | -0.005 | $H_2$ | Not Supported |
| FLS → CEX | 0.306 | 0.295 | $H_3$ | Supported |
| CSA → CLY | 0.276 | 0.410 | $H_4$ | Supported |
| CEX → CLY | 0.269 | 0.314 | $H_5$ | Supported |
| CEX → CSA | 0.709 | 0.558 | $H_6$ | Supported |

$^*p < 0.05$.

state can be thought of an escape from this daily routine and therefore, is being valued more by today's consumers. The results signify that investment to provide consumers with a seamless experience that makes them feel in the flow not only helps the brands have their customers entertained, but this seamless experience does also help improve brand loyalty and customer satisfaction in the long run.

As for the customer loyalty dimension, the results of the study can be said to conform with the results as found by Shim, Forsythe, Kwon [13]. However, this paper also makes a significant contribution to the literature by studying and validating a positive relationship between the online flow state and customer satisfaction (in addition to customer loyalty) through brand experience.

## 7. Conclusion

Triggered by the covid-19 pandemic, e-commerce adoption rates accelerated even more due to the increased adoption of online medium. During these tough times, it has become more important than ever for e-businesses to provide seamless experiences and build long-lasting, profitable relationships with their customers.

The results of the present study demonstrate that online flow state as perceived by customer trust of university students in Turkey influences their satisfaction and loyalty towards e-commerce websites indirectly through customer experience.

The online flow state aims at creating a totally absorbing, engaging experience with a brands' website. Naturally, this experience is expected to happen momentarily.

Therefore, it is not expected to affect long-term phenomena such as customer loyalty and customer satisfaction considering theoretically. Although online streaming has been studied from different angles, it cannot be said that the number of studies investigating the effect of online streaming on total brand experience, customer loyalty and customer satisfaction is not high in the current literature, except for a study by Shim, Forsythe, Kwon [13]. However, that study investigated only the flow- brand experience- loyalty relationships.

This study has aimed at adding to the extant research by incorporating customer satisfaction dimension and analyzing the direct and indirect effect of all of those construct's phenomenon together using structured equation modeling.

The findings of the present study have important managerial implications. The results imply that fostering a flow state shouldn't be viewed as a nice-to-have feature in an e-commerce setting. But rather, doing so does help the e-commerce brands achieve their longer-term objectives. The moment a customer enters an online platform for a product, the amount of time he / she spends there and the pleasure derived all have critical importance with regard to customer satisfaction and customer loyalty. During the flow state, customized and personalized offerings, influential visual designs on the online platforms yield a more important

customer experience than purchasing experience in facilitating the shopping process and rendering it an enjoyable one. In sum, the websites that keep the customer in the flow will be preferred more.

## Supporting information

**S1 Data.**
(SAV)

## Author Contributions

**Conceptualization:** Adnan Veysel Ertemel, Mustafa Emre Civelek.

**Data curation:** Mustafa Emre Civelek, Murat Çemberci.

**Formal analysis:** Mustafa Emre Civelek.

**Investigation:** Güzide Öncü Eroğlu Pektaş.

**Methodology:** Mustafa Emre Civelek, Murat Çemberci.

**Project administration:** Adnan Veysel Ertemel.

**Resources:** Adnan Veysel Ertemel, Mustafa Emre Civelek, Güzide Öncü Eroğlu Pektaş, Murat Çemberci.

**Software:** Mustafa Emre Civelek.

**Supervision:** Adnan Veysel Ertemel, Murat Çemberci.

**Validation:** Mustafa Emre Civelek, Murat Çemberci.

**Visualization:** Mustafa Emre Civelek.

**Writing – original draft:** Adnan Veysel Ertemel, Mustafa Emre Civelek, Güzide Öncü Eroğlu Pektaş.

**Writing – review & editing:** Adnan Veysel Ertemel, Güzide Öncü Eroğlu Pektaş.

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
