## [Decision Letter · Decision Letter 0]

20 May 2021

PONE-D-21-08998

The Role of Customer Experience in the Effect of Online Flow State on Customer Loyalty

PLOS ONE

Dear Authors,

Thank you for submitting your manuscript to PLOS ONE. After careful consideration, we feel that it has merit but does not fully meet PLOS ONE’s publication criteria as it currently stands. Therefore, we invite you to submit a revised version of the manuscript that addresses the points raised during the review process.

Please see comments below.

We look forward to receiving your revised manuscript.

Kind regards,

Dejan Dragan, PhD

Academic Editor

PLOS ONE

Journal Requirements:

2. Thank you for submitting the above manuscript to PLOS ONE. During our internal evaluation of the manuscript, we found significant text overlap between your submission and the following previously published works.

- https://doi.org/10.1108/IJBM-10-2014-0139

- https://coek.info/pdf-the-impact-of-e-service-quality-and-customer-satisfaction-on-customer-behavior-i.html

- https://doi.org/10.1016/j.jretconser.2017.07.010

- http://www.jecr.org/sites/default/files/16_1_p04.pdf

We would like to make you aware that copying extracts from previous publications, especially outside the methods section, word-for-word is unacceptable, even for works which you authored. In addition, the reproduction of text from published reports has implications for the copyright that may apply to the publications.

Please revise the manuscript to rephrase the duplicated text, cite your sources, and provide details as to how the current manuscript advances on previous work. Please note that further consideration is dependent on the submission of a manuscript that addresses these concerns about the overlap in text with published work.

4. We note you have included a table to which you do not refer in the text of your manuscript. Please ensure that you refer to Table 3 in your text; if accepted, production will need this reference to link the reader to the Table.

Additional Editor Comments:

The comments of the reviewers are diverse, from "minor revision" all the way to rejection of the paper. I suggest the authors to strictly follow all the reviewers' comments. AE DD

Reviewers' comments:

Reviewer's Responses to Questions

**Comments to the Author**

1. Is the manuscript technically sound, and do the data support the conclusions?

Reviewer #1: Yes

Reviewer #2: Partly

Reviewer #3: Partly

Reviewer #4: Partly

2. Has the statistical analysis been performed appropriately and rigorously? 

Reviewer #1: Yes

Reviewer #2: I Don't Know

Reviewer #3: No

Reviewer #4: Yes

3. Have the authors made all data underlying the findings in their manuscript fully available?

Reviewer #1: Yes

Reviewer #2: No

Reviewer #3: No

Reviewer #4: No

4. Is the manuscript presented in an intelligible fashion and written in standard English?

Reviewer #1: Yes

Reviewer #2: Yes

Reviewer #3: Yes

Reviewer #4: No

5. Review Comments to the Author

Reviewer #1: The manuscript was presented in an understandable fashion and written in standard English. It is well structured technically. The purposeful division of the material is to make it easier for the reader to understand the terms and easily find the issues of interest. The advantage of the article is the order of the content - the principle "from general to specific" was used here. The research hypotheses are well formulated and the findings provide a basis for answering them. Also noteworthy is the description of the research - clear and logical. It is evident that the authors understand the subject, pose the right questions and are able to answer them. The statistical analysis has been performed appropriately: quantitative data were collected by means of the questionnaire designed in a five-point Likert scale. Firstly, confirmatory faction analyses and reliability analysis were conducted to determine the validity and reliability of the scale. SEM as a multi variable statistical technique was employed to test the hypotheses. This technique was used to understand the indirect and direct effects in the theoretical model and to decrease measurement errors. The analyses were performed with SPSS and AMOS statistics programs. The authors also provided the necessary data on which the manuscript's conclusions are based. It is also worth noting the extensive literature that the authors used as a basis for preparing the article. The authors concluded their argument with a discussion and a well-designed conclusion. To sum up, this study makes a significant contribution to the literature by highlighting this relationship in online context in e-commerce environment. In addition to the theoretical layer, the article can provide application value for managers. The results indicate that investment to provide consumers with a seamless experience that makes them feel in the flow not only helps the brands have their customers entertained, but this seamless experience does also help improve brand loyalty and customer satisfaction in the long run.

Reviewer #2: The topic is interesting, however, I am sorry to say that I found little merit in this paper. The writing of this paper is quite superficial, and the proposed theory (flow theory) is not well-elaborated. In the introduction, the authors do not provide convincing arguments regarding the significance of this study, and they made insufficient effort to explain the rational of this study. In the literature section, the writing is quite descriptive and lacks the connection with the context of this study (section 2.1, 2.1.1, 2.2). In addition, the selected literatures are outdated, there are definitely some more recent studies on this topic. The data analysis seems reasonable, however, I don’t think these findings can bring much new knowledge to the field of e-commerce.

Reviewer #3: We think that article takes into consideration an interesting topic, that could have a special importance nowadays taking account of the profound transformations of consumption models under the pressure of COVID-19 pandemia. Online consumption becomes more and more present for larger mass of consumers and all the variables that can describe this phenomena deserves to be investigated as deeply as possible.

In order to achieve a proper level required for publication we strongly encouraged authors to take account off the following remarks/ requests:

1. Authors have to complete a supplementary review on English language in order to eliminate some minor typing errors. For example at line 57....”He brands now have......it should be: ..”The brand now have....

Also in line 187...” can build commitment in terms of”.....

Line 319, please correct the error:....”Firstly, confirmatory faction analyses”....

and so on...

2. The sample was reached using online communication, being targeted students, but authors should specify from what universities? Are there from a single city like Istanbul or are from different universities across Turkey?

Also it is advisable to present the structure of the sample (age, income etc - other descriptive variables used). In order to have a proper description of the sample authors can find a model within the following article: https://sciendo.com/article/10.2478/mmcks-2020-0031, at page 541.

3. Who is the sample representative of? students in general? Students only from the capital Istanbul?

4. As regarding the model, in order to have a proper explained methodology it is advisable to take into consideration more indicators / statistical tests (for example Kaiser–Meyer–Olkin (KMO) test etc). Authors can find a proper analysis and presentation for a similar model within the following article: https://www.mdpi.com/2071-1050/12/22/9780/pdf at page 8. The example also indicates the statistical values which are relevant for hypothesis validation.

5. Where does the information on value 3 related to “χ2 / DF” come from? See lines 351-352. Why are these indicators (referring to IFC, IFI, RMSEA) different at line 353 from the values at line 403 (and below lines 410-411)? What exactly does the note on line 403 refer to? Where should the note be placed? Under figure 2? The authors need to explain this passage more clearly.

6. Table 2 must be redone as a design - columns made wider to allow the text to be framed on a single line - the current form denotes negligence in terms of appearance for the finished version of the article.

Given that the Cronbach-Alpha values are not above 9 (which would have meant the immediate validation of the model), the table must be completed with the values of other coefficients (see the model for the article indicated above) that explicitly show this thing. What are the explicit values for p -value (Sig) for each of the 4 variables considered separately.

7. Figure 2, the one referring to the coefficients of the model, needs to be redone, in order to be more visible all the details of the figure - possibly it is recommended to import from SPSS or AMOS.

8. The model needs to be explained more clearly. What exactly do the notations in the model mean and what are the values for each item that are behind the four variables of the model?

9. The bibliography is old. Articles from the last 5 years should be considered mainly

10. To verify that the citations are unitary and fit into the policy of the journal.

To correct errors related to magazine names, for example line 550 - 559, all magazine names must be entered with

large initials, in a unitary way.

Avoid lack of rigor in this regard - for example - lines 645, 647, 648, 707, 754 (pages, volume, etc.) The references are a very important part of the article, being a subject for article rejection in many occasions.

Reviewer #4: I congrats the authors for the present study and I wish that the suggestions presented here might allow the authors to show the manuscritpt's full potential in the next opportunity.

Some suggestions:

1. Submit the paper to a English proofreading review.. Ex: Line 57 - "he brands..""

2. Introduction: a) The central objective of the paper is not clear stated in the introduction; b) the second paragraph seems to be overlapped by the third which is more well based on citation. Therefore, the introduction lacks of contextual facts to sustain the problem that the paper proposes to tackle. The authors assert that online flow is a " popular concept studied in various contexts in the literature", but it's not sufficiently proven the reasoning in the introduction neither in the literature review.

3. Conceptual background: a) I suggest that the mention of the usage of SEM method should be made at the methodology section, b) also, the theoretical model figure should presented in the beginning of section 3. C) in this section I suggest the authors to present a Table 1 summarizing prior studies regarding online flow studies organized by some criteria such as constructs and casual relations tested, methods, researched public, country, main findings. The final line of the table should be your paper.

4. research methods. A) I suggest to include the questionnaire afirmations in table 1. B) inform whether a pre teste were carried out; c) clarify if the questionnaire was apllied in English or Turkish language;

4) Discussion – the authors affirms twice “study makes a significant contribution to the literature” (lines 442 and 465) but these affirmation lacks of arguments based on prior reviewed literature.

5) Conclusion – The covid19 scenario was brough to the discussion in conclusion. Why was not discussed either in the introduction? I suggest also to enhance the managerial contributions in a separate subsection of the conclusion and also add “research limitation and future studies”.

6. PLOS authors have the option to publish the peer review history of their article (what does this mean?). If published, this will include your full peer review and any attached files.

Reviewer #1: **Yes: **Anna Dziadkiewicz

Reviewer #2: No

Reviewer #3: No

Reviewer #4: No

---

## [Author Response · Author response to Decision Letter 0]

24 Jun 2021

Dear Editor,

Thank you for your comments on the manuscript.

The manuscript has been revised in the light of reviewer comments.

1. Conformance to PLOS ONE’s style requirements

The manuscript has been revised to conform to the PLOS ONE’s style requirements. The changes have been highlighted in the manuscript with track changes open in the Word document. 

2. Text overlap with 4 sources

The part of the text related with the mentioned 4 sources has been rephrased and the sources has been mentioned in the text.

3. Availability of the Research Dataset

We have uploaded the research dataset for your information. 

4. Missing reference to Table 3 within the text

Additional sentence is put to reference Table 3.

Reviewer Comments:

Reviewer #2: 

Significance of the study

New text has been added to the Introduction and Conceptual background parts to underline the significance of the study.

Reviewer #3:

1. Typing errors

The whole manuscript has been revised again to correct the typing errors including but not limited to the mentioned ones. The corrections have been highlighted with track changes on in the Word document.

2-3 Sampling and descriptive statistics

The following paragraph has been added to 4.1 Measures and Sampling part which includes answers to the raised questions.

The questionnaire was distributed to more than 700 individuals residing in Turkey via an online from. The survey was conducted in Turkish, among university students residing in 7 biggest cities in all geographical regions in Turkey, namely; İstanbul, İzmir, Antalya, Samsun, Ankara, Erzurum, Diyarbakır. The online questionnaire was accepted only from those adults with their explicit consent and who had previously purchased online in the past 12 months. 538 valid questionnaires from individuals were collected. 334 of the respondents were female and the remaining 204 respondents were male. 

4-5-6-7-8 Authors Comments to the criticisms on Validity

The method of the article is very robust and the determination of validity is reported correctly. In a theoretically determined model, construct validity refers to convergence of observed variables that are connected to the same latent variable (convergent validity) and dissociation of observed variables from other observed variables that are connected to other latent variables (discriminant validity). The construct validity indicates that the observed variables do not measure any latent variable other than they connected in the conceptual model. But in this case, it would not be correct to say that the validity of the construct is fully realized without confirming the reliability of the scale (Gerbing & Anderson, 1988). 

Convergent validity indicates that the correlations between questions constituting a construct are high. In structural equation modeling method, it is necessary to look at the results of confirmatory factor analysis to determine the convergent validity of the scales used to measure the dimensions constituting the conceptual model of the research. The measurement model part of structural equation models corresponds to confirmatory factor analysis (Confirmatory Factor Analysis - CFA). Therefore, if the measurement model fit indices are low, there is no need to test the structural model (See also Figure 1. Demarcation between Measurement Model and Structural Model). Because the scales used to measure the dimensions that make up the conceptual model will not be validated. Therefore, if the measurement model is insufficient, the fit indices of the structural model will be low. The t test results of all the coefficients in the measurement model should indicate that the coefficient values are different from zero. The standard value of each coefficient in the measurement model is the factor loadings of the confirmatory factor analysis. Each factor load should be higher than 0.50. Otherwise, the fit indices of the general model will be adversely affected. The fact that the factor loads are above 0,5 is evidence of convergent validity. If the critical rate value of a question in CFA results is greater than 2 as an absolute value this means that this item is loaded to the factor it is connected. 

Discriminant validity is the measure of the level at which a structure in a measurement model differs from other structures. It is an indicator of a low correlation between the questions that form a construct and other questions that form another construct. To find the discriminant validity for each dimension, we first need to calculate the Average Variance Extracted (AVE) value for each dimension. The acceptable AVE value must be greater than 0.50 or 0.50. However, as noted in the previous section, this value confirms convergent validity when examined alone (Fornell & Larcker, 1981). In order to determine discriminant validity, it is also desirable that the values of the AVE for each construct in the data set are larger than the correlation coefficients of that construct with the other constructs. In this case, it can be determined that the scales used have discriminant validity for each dimension. AVE value alone does not indicate discriminant validity but the square root of the AVE value of each construct is larger than the inter-dimensional correlation value it can be said that there is discriminant validity (Fornell & Larcker, 1981).

9. Bibliography

The extant literature has been revised extensively including but not limited to the recent literature.

10. Reference names

References section has been revised thoroughly to correct any mistakes.

Reviewer #4

Typing errors have been corrected. The corrections are highlighted with track changes open.

The second paragraph in the introduction section has been removed and the third paragraph has been revised.

New sentences were added to discuss the Covid-19 scenario also in the introduction. 

Note to the reviewer: No pre-test was carried out in the research.

---

## [Editor Report · Decision Letter 1]

2 Jul 2021

The Role of Customer Experience in the Effect of Online Flow State on Customer Loyalty

PONE-D-21-08998R1

Dear Authors,

We’re pleased to inform you that your manuscript has been judged scientifically suitable for publication and will be formally accepted for publication once it meets all outstanding technical requirements.

Kind regards,

Dejan Dragan, PhD

Academic Editor

PLOS ONE

Additional Editor Comments (optional):

All reviews and comments have been appropriately considered. Accordingly, I recommend the acceptance of the paper. AE DD.
---

## [Editor Report · Acceptance letter]

6 Jul 2021

PONE-D-21-08998R1 

The Role of Customer Experience in the Effect of Online Flow State on Customer Loyalty 

Dear Dr. Eroğlu Pektaş:

I'm pleased to inform you that your manuscript has been deemed suitable for publication in PLOS ONE. Congratulations! Your manuscript is now with our production department. 

Kind regards, 

on behalf of

Dr. Dejan Dragan 

Academic Editor

PLOS ONE